# Trends in Anti-Tumor Effects of *Pseudomonas aeruginosa* Mannose-Sensitive-Hemagglutinin (PA-MSHA): An Overview of Positive and Negative Effects

**DOI:** 10.3390/cancers16030524

**Published:** 2024-01-25

**Authors:** Dragica Bozic, Jovana Živanović, Katarina Živančević, Katarina Baralić, Danijela Đukić-Ćosić

**Affiliations:** 1Department of Toxicology “Akademik Danilo Soldatović”, Faculty of Pharmacy, University of Belgrade, Vojvode Stepe 450, 11221 Belgrade, Serbia; jovana.zivanovic@pharmacy.bg.ac.rs (J.Ž.); katarina.zivancevic@pharmacy.bg.ac.rs (K.Ž.); katarina.baralic@pharmacy.bg.ac.rs (K.B.); danijela.djukic.cosic@pharmacy.bg.ac.rs (D.Đ.-Ć.); 2Center for Laser Microscopy, Faculty of Biology, Institute of Physiology and Biochemistry “Ivan Djaja”, University of Belgrade, Studentski trg 16, 11158 Belgrade, Serbia; 3Center for Toxicological Risk Assessment, Faculty of Pharmacy, University of Belgrade, Vojvode Stepe 450, 11221 Belgrade, Serbia

**Keywords:** cancer, PA-MSHA, apoptosis, immunomodulating activity, safety

## Abstract

**Simple Summary:**

Finding the best treatment for cancer remains one of the biggest scientific challenges in the 21st century. Available cancer treatments have issues like drug resistance and severe side effects, indicating the need for alternative therapies, especially for patients who do not respond well to regular methods like chemotherapy or surgery. Recently, a new approach, bacteria-mediated cancer therapy, has gained attention. A specific bacterium called *Pseudomonas aeruginosa mannose-sensitive-hemagglutinin* (PA-MSHA) has shown promise in inhibiting the growth of various cancers, like breast and lung cancer. It works by halting tumor cell growth and prompting their natural death. Clinical studies suggest that PA-MSHA can boost the effectiveness of chemotherapy and activate the immune system against cancer, with mild side effects like fever and skin irritation. This review aims to summarize the current knowledge about the usage of PA-MSHA for cancer treatment, from laboratory studies to clinical trials, understanding both its benefits and potential drawbacks.

**Abstract:**

Cancer is a leading cause of death worldwide, for which finding the optimal therapy remains an ongoing challenge. Drug resistance, toxic side effects, and a lack of specificity pose significant difficulties in traditional cancer treatments, leading to suboptimal clinical outcomes and high mortality rates among cancer patients. The need for alternative therapies is crucial, especially for those resistant to conventional methods like chemotherapy and radiotherapy or for patients where surgery is not possible. Over the past decade, a novel approach known as bacteria-mediated cancer therapy has emerged, offering potential solutions to the limitations of conventional treatments. An increasing number of in vitro and in vivo studies suggest that the subtype of highly virulent *Pseudomonas aeruginosa* bacterium called *Pseudomonas aeruginosa mannose-sensitive-hemagglutinin* (PA-MSHA) can successfully inhibit the progression of various cancer types, such as breast, lung, and bladder cancer, as well as hepatocellular carcinoma. PA-MSHA inhibits the growth and proliferation of tumor cells and induces their apoptosis. Proposed mechanisms of action include cell-cycle arrest and activation of pro-apoptotic pathways regulated by caspase-9 and caspase-3. Moreover, clinical studies have shown that PA-MSHA improved the effectiveness of chemotherapy and promoted the activation of the immune response in cancer patients without causing severe side effects. Reported adverse reactions were fever, skin irritation, and pain, attributed to the overactivation of the immune response. This review aims to summarize the current knowledge obtained from in vitro, in vivo, and clinical studies available at PubMed, Google Scholar, and ClinicalTrials.gov regarding the use of PA-MSHA in cancer treatment in order to further elucidate its pharmacological and toxicological properties.

## 1. Introduction

Cancer is a complex and multifactor disease that affects millions of people worldwide. The impact of cancer on individuals and society cannot be overstated, as it causes significant morbidity, mortality, and economic burden. Therefore, great efforts are being made to find effective therapeutic strategies and ways to prevent tumors from becoming resistant to therapy [1,2,3,4]. It is characterized by an uncontrollable proliferation and invasive growth of altered cells in the body, cell resistance to death, insensitivity to growth suppressors, sustained angiogenesis, and the ability to metastasize [5,6]. Conventional cancer treatment includes surgery and tumor resection, radiotherapy, and chemotherapy [6]. The main cancer therapy targets are tubulin protein, cyclin-dependent kinases (CIKs), epidermal growth factor receptor (EGFR), Ras protein, and, in recent years, cancer stem cells (CSC) [7,8]. Tubulin, cyclin-dependent kinases, and the Ras protein regulate the cell cycle of cancer cells and allow continuous proliferation and growth of tumors. Therefore, tubulin and CIK inhibitors have emerged as important anti-cancer molecules that stimulate apoptosis and stop cancer progression [9,10]. The EGFR was shown to be overexpressed in different cancer forms. Its expression positively correlates with cancer progression and poor prognosis, which led to the development of numerous EGFR inhibitors, such as afatinib, brigatinib, or vandetanib [11]. Finally, it destroys CSC or tumor-initiating cells by targeting their surface markers, such as CD133, CD44, and CD24. Antibodies directed against these markers prevent tumor-induced immune inhibition and prolong the activity of effector T cells, which execute tumor cell apoptosis [7].

However, it is challenging to find the optimal treatment due to the disease’s multifactorial physiology, drug resistance, occurrence of undesirable toxic effects, and non-selectivity of the drugs [12]. Therefore, researchers are investigating and developing new, complementary treatment strategies to target tumors and cancerous tissues more efficiently [13].

In the last 10 years, bacteria-mediated cancer therapy has emerged as a novel antineoplastic strategy that has the potential to overcome some limitations of conventional cancer therapies [13,14]. It is known that bacteria exhibit anti-tumor effects by activating the immune system via different mechanisms, such as upregulation of reactive oxygen species, which further promote inflammatory responses or stimulation of effector T cells that have tumoricidal activity [15]. Studies have shown that certain bacterial strains, such as *Salmonella typhimurium, Streptococcus pyogenes*, and *Pseudomonas aeruginosa*, have high anti-tumor virulence [16]. The use of bacteria for cancer treatment has shown promising results in a variety of cancer cases. Moreover, the low toxicity of bacteria and their products, along with their limited side effects on healthy cells, provides further hope for the use of bacteria as adjuvant chemotherapeutics [17,18]. Various bacterial species, forms, and bacterial metabolites have been utilized to selectively multiply, influence tumors, and restrain their growth [17]. Anticancer bacteria-based immunotherapy methods could use bacteria in living, attenuated conditions, or in genetically engineered forms. Additionally, spores of anaerobic bacteria can be used to synthesize, develop, and produce anticancer agents, or they can be utilized as carriers for gene and drug delivery to tumor tissues [19]. Bacterial spores can exert their anticancer role if they encounter hypoxic-necrotic tissues, where they are able to germinate and multiply. Moreover, in the middle of the 20th century, researchers showed that genetically modified bacteria can selectively destroy tumors, alter cellular enzyme expression, and consequently lead to a successful outcome [18]. The most studied tumor-targeting bacteria are engineered strains of *Salmonella*, *Clostridium*, and *Listeria*. Nevertheless, a growing number of modified bacteria are currently under investigation for their potential as anti-cancer therapeutics, with *Pseudomonas aeruginosa* being one notable example [20].

*Pseudomonas aeruginosa* is a gram-negative, aerobic bacterium that is innocuous to healthy individuals but can cause serious infections in patients with weakened immune systems [21]. Despite the fact that *Pseudomonas aeruginosa* is classified as an aerobe, it functions as a facultative anaerobe, capable of producing energy under oxygen-limited conditions by using alternative electron acceptors such as nitrate (NO_3_^−^), nitrite (NO_2_^−^), and nitrous oxide (N_2_O) [22]. While a live form of *Pseudomonas aeruginosa* is highly virulent and can cause fatal infections in humans, its inactivated, genetically modified form and its toxin, exotoxin A, have been proposed as potential immune-modulating and cytotoxic anti-tumor agents [23,24,25]. A genetically transformed subtype of *Pseudomonas aeruginosa* bacteria is called *Pseudomonas aeruginosa-mannose-sensitive hemagglutinin* (PA-MSHA), a bacteria killed at temperatures above 65 °C. It is a biologically engineered strain coated with fragile and straight MSHA (mannose-sensitive hemagglutination) fimbriae [26]. The MSHA surface decreases the toxic potential of *Pseudomonas aeruginosa* by preventing the expression of more harmful virulence factors on its surface [23]. In vivo and clinical data show that PA-MSHA can stimulate the immune response via the TLR4 signaling pathway and thus prevent the development of sepsis after surgery. PA-MSHA stimulated the release of serum pro-inflammatory mediators (TNF-α, IL-1β, and IL-6) 8 days after the intervention but increased anti-inflammatory mediators (IL-4 and IL-10) after 16 days, balancing the activity of the immune system to prevent infections [23,27]. PA-MSHAs immune-related and anti-tumor characteristics led to its approval in China in 1998 as a complementary cancer treatment that aims to boost patients’ immune systems and thus improve the treatment of malignant tumors such as breast, gastric, bladder, hepatocellular, and colorectal [16,28,29].

PA-MSHA has demonstrated positive anti-cancer effects across various cancer types by interacting with the immune system in several ways: it induces M1 polarization of macrophages, Th1 immune response promotion, and increases the secretion of IFN-γ while reducing the production of IL-10 and IL-4 [30,31].

However, it has been observed that the application of PA-MSHA can cause side effects such as fever, skin irritation, and/or rashes, which occurred in patients with Her-2 negative metastatic breast cancer after combined treatment with anthracycline, taxol, and PA-MSHA [32]. These effects could be linked with PA-MSHA-induced release of pro-inflammatory cytokines and immune stimulation [33].

Therefore, this review aims to investigate and explain the current implications of PA-MSHA in cancer treatment by examining the results from in vitro, in vivo, and clinical studies, as well as to enhance our understanding of its pharmacological and toxicological properties. To the best of our knowledge, this is the first review to provide an in-depth examination of the positive and negative anti-tumor properties of PA-MSHA.

## 2. Anti-Tumor Potential of PA-MSHA: In Vitro Results

PA-MSHA has been shown to inhibit the growth of cancer cells in numerous in vitro studies. In their recent work, Zhao et al., (2016) explained that PA-MSHA stimulates apoptosis and cell cycle arrest in different non-small cell lung cancer cells (PC-9, A549, and NCI-H1975) in a time- and dose-dependent manner. The NCI-H1975 was more sensitive to PA-MSHA than PC-9 and A549, with an IC_50_ of 0.140 × 10^9^/mL at 72 h. For the inhibition of 50% of A549 cells, more PA-MSHA was required, leading to a final IC_50_ of 0.922 × 10^9^/mL at 72 h. Moreover, mechanistic studies revealed that PA-MSHA increases the level of activated caspase 3 and caspase 9 and consequently stimulates intrinsic apoptotic pathways [16]. Caspases are known for their role in the regulation of programmed cell death, where caspase-9 initiates the process while caspase-3 contributes to the execution of apoptosis [34]. When activated, caspase-3 governs tumor cytotoxicity, indicating the anti-cancerogenic potential of caspase-3 activators [35,36], including PA-MSHA. Additionally, in a study performed by Zhao et al., (2016), the combinational treatment of gefitinib and PA-MSHA was tested for the treatment of non-small cell lung cancer both in vitro and in vivo. The authors found that combinational treatment was more effective in inhibiting proliferation and stimulating apoptosis of cancer cells than either agent alone. Expectedly, the PA-MSHA and Gefitinib combination resulted in increased caspase-3/caspase-9 cleavage and inhibition of EGFR-dependent activation of the AKT/ERK pathway involved in cancer cell survival [37]. As mentioned, inhibition of EGFR is a known mechanism of specific tyrosine kinase inhibitors, or EGFR antagonists, commonly used in metastatic epithelial cancers [38]. The suppression of EGFR-regulated RAS-RAF-MAPK and PI3K-AKT-signaling pathways results in a reduction in cancer cell proliferation and increased cancer cell death [39]. It was found that PA-MSHA inactivates EGFR signaling in bladder, pancreatic, and breast cancer cells and in vivo in tumor-bearing mice [40,41,42]. Interestingly, Liu et al., (2010) found that PA-MSHA-mediated inhibition of EGFR was mannose-dependent [42].

Moreover, PA-MSHA inhibited the proliferation of human breast cancer cell lines and stimulated their apoptosis. Flow cytometry analysis revealed that PA-MSHA-induced cell cycle arrest in the G0/G1 phase prevented breast cancer cell multiplication. Additionally, amplified expression of cleaved caspase 3, 8, 9, and Fas protein indicated a higher rate of apoptosis in breast cancer cells treated with PA-MSHA in comparison to control [43]. In the next study, researchers investigated if PA-MSHA could reduce the growth of doxorubicin-resistant breast cancer cells. They treated doxorubicin-resistant and non-resistant cells, MCF-7/ADR and MCF-7, respectively, with PA-MSHA for 48 h. Interestingly, PA-MSHA inhibited growth and induced apoptosis of MCF-7/ADR cells but not MCF-7 cells, with an IC_50_ of 1.421 × 10^9^ cells/mL at 12 h and an IC_50_ of 0.848 × 10^9^ cells/mL at 24 h. The authors concluded that PA-MSHA can be an effective treatment for doxorubicin-resistant breast cancer. The underlying mechanism can be at least partly explained by the suppression of Nrf2 and p62 [44]. The Nrf2 transcriptional factor and p62 protein play a significant role in autophagy, a metabolic process that helps cancer cells in the late stages of development survive in nutrient-depleted environments. The autophagy adaptor p62 interacts with Keap1, a Nrf2 inhibitory protein, leading to increased stabilization and transcriptional activity of Nrf2. Nrf2 further helps cancer cells overcome stress conditions [45]. Therefore, by inhibiting Nrf2 and p62, PA-MSHA prevents the adaptation of cancer cells to a shortage of nutrients and promotes apoptosis.

Yin et al., (2016) exposed human cervical cancer cell lines to 0.15, 0.3, or 0.6 × 10^9^/mL PA-MSHA for 24 h, 48 h, and 72 h and found time- and dose-dependent anti-tumor potential. The inhibition of tumor cell proliferation was highest after 72 h, while the most effective dose was 0.6 × 10^9^/mL. Further analysis revealed that activation of phosphatase and tensin homolog (PTEN) and consequential inhibition of the p-AKT pathway are responsible for PA-MSHA-induced inhibition of cancer cell proliferation and invasion [46]. Phosphatase and tensin homolog are the natural inhibitors of the PI3K/AKT pathway, and their loss or inactivation has been recorded in various cancer types, including ovarian, breast, urothelial, and prostate cancer [47]. Moreover, PTEN is a potent tumor suppressor, and its stimulation or reactivation has been proposed as a potential anti-tumor therapeutic strategy [48]. Similar results were seen in bladder cancer cells: PA-MSHA induced time- and concentration-dependent inhibition of tumor cell growth by stimulating protein expression of cleaved caspase-8 and -9, as well as Fas protein. Additionally, cells treated with PA-MSHA experienced downregulation of the PI3K/AKT pathway [49].

Furthermore, PA-MSHA was able to arrest human hepatocellular carcinoma cells in the S phase of the cell cycle and, thus, promote their apoptosis. This effect was mediated by mannose residues on the bacterial surface, which directly activated caspase-8 and the extrinsic apoptotic pathway [50]. Additionally, Wei et al., (2023) investigated the impact of PA-MSHA on programmed cell death ligand, PD-L1, expression in hepatocellular carcinoma cells. They found that PA-MSHA treatment reduced the transcription of the PD-L1 gene and, consequently, suppressed the expression of the PD-L1 molecule on the cancer cell surface [51]. High PD-L1 expression on tumor cells was linked to tumor immune escape and more pronounced tumor growth. The PD-L1 ligand reacts with PD-1 to inhibit T cell activation, leading to the inhibition of the immune response [52]. Similarly, Zhang et al., (2014) showed that the anti-tumor effects of PA-MSHA could be linked to the activation of immune cells, mainly cytokine-induced killer (CIK) cells, and the downregulation of inhibitory cell surface markers, such as PD-1 and CTLA-4, expressed on effector T cells [53]. PA-MSHA also promoted activation of dendritic cells and, thus, stimulation of the Th1-immune response, leading to increased secretion of the pro-inflammatory cytokine IFNγ and decreased secretion of the immune-suppressors interleukin-10 (IL-10) and interleukin-4 (IL-4). Importantly, this effect was observed only when concentrations of up to 1.15 × 10^9^ mL^−1^ were added to the medium with dendritic cells, while higher concentrations decreased dendritic cell activity [54]. Taken together, these findings suggest that PA-MSHA has the capacity to activate the immune response both through direct interaction with immune cells and indirectly by halting the immune inhibition induced by tumors.

Therefore, based on the previously mentioned in vitro results, it may be suggested that PA-MSHA contributes to tumor suppression by: (1) stimulation of cancer cell apoptosis; (2) inhibition of cancer cell proliferation; and (3) reactivation of immune cells and stimulation of cytotoxic immune response (Figure 1).

## 3. Anti-Tumor Potential of PA-MSHA: In Vivo Results on Animal Models

The effectiveness of PA-MSHA against tumors has been confirmed in numerous animal models as well (Table 1). Conducted in vivo studies have shown anti-tumor properties in breast, bladder, and lung cancer when PA-MSHA is used alone or in combination with chemotherapy [3,40,42]. For example, it was also shown that subcutaneous administration of PA-MSHA (2.2 × 10^10^ cells/mL) for 45 days reduces the growth and development of breast cancer with EGFR overexpression [42]. Similarly, in a study on a mouse model of pancreatic carcinoma with enhanced expression of the EGFR factor, subcutaneous administration of PA-MSHA (2.2 × 10^10^ cells/mL) in combination with paclitaxel (10 mg/kg, 2 times a week, for 7 weeks) led to cancerous cell apoptosis and growth inhibition [41]. This review will further provide a comprehensive examination of the anti-tumor properties of PA-MSHA observed in animal models, offering a detailed overview.

The anti-tumor potential of PA-MSHA was also seen in bladder cancer [5,55,56,57]. This bacterium inhibited tumor growth, reduced invasiveness, and promoted tumor cell apoptosis, mainly by activating the transcription factor NF-kB, which is necessary for the synthesis of proteins involved in the tumor cell cycle. Additionally, PA-MSHA increased the ratio of M1 to M2 macrophages, which leads to the stimulation of an inflammatory response in the tumor microenvironment and, consequently, a reduction in tumor size. More specifically, PA-MSHA promoted the formation of M1-like macrophages, which serve as tumor-suppressor macrophages, and repressed the production of M2-like macrophages, which promote angiogenesis and tumor growth refs. [55,56]. Similarly, Li et al., (2015) described the immunostimulatory potential of PA-MSHA in a mouse model bearing a tumor. They found that mice treated with PA-MSHA had higher levels of effector T cells and pro-inflammatory cytokines, such as IFN-γ and TNF-α, which have an anti-tumor effect [57]. Also, in the study conducted by Hung et al., (2022), PA-MSHA showed the ability to activate T-cells, highlighting its potential to sensitize refractory “cold” tumors to immunotherapy. Thus, PA-MSHA, when administered together with the anti-programmed cell death antibody, anti-PD1-antibody, was able to suppress tumor growth and prolong the survival of tumor-bearing mice [55]. Another anti-tumor PA-MSHA-induced mechanism was proposed by Chang et al., (2014). They suggested that these effects were achieved by inhibiting the EGFR signaling pathway and the consequent activation of the apoptotic process. Of note, PA-MSHA induced apoptosis of tumor cells at a higher dose and prolonged exposure compared to healthy uroepithelial cells. This suggests that the effects of PA-MSHA may be specifically directed at tumor cells [5].

Zhang et al., (2022) demonstrated the efficacy of PA-MSHA in the treatment of colorectal cancer resistant to cetuximab. As expected, in tumor-bearing mice treated with PA-MSHA, the size of the tumor was reduced while the survival of the animals was extended. The miR-7-5p/Akt/Wnt-β-catenin signaling pathway was proposed as the main target of PA-MSHA responsible for the inhibition of tumor invasion, migration, and induction of tumor cell apoptosis. The authors explained that cetuximab-resistant colorectal carcinoma exhibits low levels of miR-7-5p expression, a microRNA identified as a suppressor of cancer tumorigenesis. The PA-MSHA treatment enhanced miR-7-5p expression, leading to the inhibition of Akt expression, consequential inactivation of the Wnt-β-catenin pathway, and finally, apoptosis [58]. Moreover, PA-MSHA has shown protective effects on the gastrointestinal tract. The administration of PA-MSHA reduced systemic and local inflammatory responses in septic shock-induced intestinal injury. This bacterium inhibited the proinflammatory activity of the TLR-NF-kB pathway and decreased ICAM-1 and VCAM-1 expression in the intestine. These adhesive molecules play a critical role in the migration of inflammatory cells to the site of inflammation [59].

The EGFR signaling pathway plays a significant role in breast cancer development. Thus, Liu et al., (2010) investigated the impact of PA-MSHA on its activity in breast cancer cells, both in vitro and in vivo. They found that the effect of PA-MSHA on breast tumors is mannose-dependent, or, in other words, the bacterium uses sugar-binding properties of cancer cells and triggers the inactivation of the EGFR signaling pathway. As a result, increased activation of caspase-9 and caspase-8 was recorded, together with stimulation of cancer cell apoptosis [42]. Moreover, as seen in an in vitro study, PA-MSHA was able to kill doxorubicin-resistant breast cancer cells. Consistently, when Wei et al., (2016) applied PA-MSHA to mice bearing doxorubicin-resistant breast cancer, inhibition of tumor growth and prolonged survival were observed. The proposed mechanism involved effects on the EGFR signaling pathway and reduction of Nrf2 and p62 expression, genes reported to significantly correlate with tumor aggressiveness, lymph node metastasis, and 5-year survival rate [44].

Moreover, Cheng et al., (2016) demonstrated that EGFR signaling plays a role in the PA-MSHA-induced apoptosis of pancreatic tumor cells. They reported that tumor-bearing mice treated with a daily subcutaneous injection of 0.1 mL of PA-MSHA (2 × 10^10^/mL) and the combination of PA-MSHA and Nab-paclitaxel (10 mg/kg, twice a week) for 7 weeks had significantly smaller tumor volumes than control mice (PA-treated mice and PA plus Nab-paclitaxel-treated mice). They showed that PA-MSHA was able to induce activation of caspase-3 and, thus, stimulate apoptosis, alone or in combination with Nab-paclitaxel, and inhibit EGFR signaling in a dose-dependent manner [41].

Other mechanisms of PA-MSHA-induced cancer cell apoptosis have been described. For example, Xu et al., (2014) explained that PA-MSHA can stimulate endoplasmic reticulum (ER) stress and thus activate the apoptotic process [28]. However, some evidence indicates that ER stress can promote tumor growth and survival through the induction of autophagy [60]. To overcome this, Xu and colleagues applied PA-MSHA together with an autophagy inhibitor, which led to tumor suppression and better overall survival in treated animals [28]. Additionally, adding PA-MSHA to regular anti-cancer therapy can increase its effectiveness and prevent the development of resistance. Zhao et al., (2016) demonstrated that PA-MSHA was able to conquer the gefitinib resistance of non-small cell lung cancer cells. Tumor-bearing mice were treated with 1 mg/kg/day of gefitinib a few days per week alone or in combination with the daily administration of 2.2 × 10^10^ cells/mL of PA-MSHA. The authors observed that combinational therapy was more effective in suppressing tumor growth and inducing apoptosis of tumor cells. Moreover, they confirmed that the anti-cancer characteristics of PA-MSHA were EGFR-signaling-dependent [37].

Additionally, Zhang et al., (2014) showed that the anti-tumor effect of PA-MSHA is dose-dependent. Administration of 10^8^ CFU/mL PA-MSHA once weekly for 3 weeks led to inhibition of tumor growth and prolonged survival of lung tumor-bearing mice more efficiently than application of lower PA-MSHA doses. Mechanistic studies revealed that the stimulation of the innate immune response and the induction of dendritic cell maturation in a TLR4-dependent manner were crucial. These effects further influenced the activation and proliferation of effector T cells, contributing to the observed outcomes [54]. Nevertheless, Wang et al., (2022) demonstrated that PA-MSHA could stimulate the immune response and inhibit immune-suppressive molecules in non-tumor-bearing mice. In contrast, tumor-bearing mice experienced an increased expression of programmed death-ligand 1 (PD-L1) protein on effector cells, ultimately resulting in treatment failure. PD-L1 is a transmembrane protein that binds to the inhibitory checkpoint molecule PD-1, transmitting an inhibitory signal to suppress the adaptive immune response. Consequently, it can be suggested that maintaining a balance between adaptive and innate immune responses is crucial for the successful treatment of PA-MSHA in tumors [61].

Moreover, PA-MSHA has demonstrated effectiveness in the treatment of hepatocellular and gastric cancer. Wang et al., (2015) showed that the efficacy of PA-MSHA in gastric cancer therapy was achieved by the activation of Nf-kB, which promotes the polarization of M1-like macrophages and increases the production of pro-inflammatory cytokines, including IL12, TNFα, and IFNγ, thereby slowing down tumor progression [31,62]. In the treatment of hepatocellular cancer, PA-MSHA inhibited the epithelial-mesenchymal transition by inhibiting NF-kB and induced apoptosis by increasing the expression of the Fas and FasL genes [63].

In conclusion, animal studies have demonstrated promising results for the utilization of PA-MSHA in the treatment of various cancer types, such as lung, bladder, breast, colorectal, gastric, hepatocellular, and pancreatic cancer. PA-MSHA has been found to be effective in inhibiting tumor growth and prolonging animal survival (Table 1), but there is limited data about its toxic potential. Available data suggests that PA-MSHA possesses finely balanced immunomodulating characteristics. However, several aspects of its mechanism of action remain unclear. For instance, the factors determining its effects on living organisms are yet to be revealed. It remains intriguing to explore whether the immune stimulation or inhibition effect is dose-dependent. Additionally, understanding whether the current state of the immune system directs PA-MSHA effects in vivo adds another layer of complexity to its study. Further research is needed to answer all of these questions, as well as to investigate the additional uses, potential side effects, and long-term effects of PA-MSHA.

**Table 1 cancers-16-00524-t001:** Anti-tumor effects of PA-MSHA: evidence from animal studies.

Animal Species	Treatment Doses	Duration	Effects	References
BALB/c nude mice	0.1 mL PA-MSHA (2.2 × 10^10^ cells/mL) s.c.	N/A	Suppressed breast tumorigenesis and the formation of metastases in the lungs	Liu et al., (2010) [42]
BALB/c nude mice	0.1 mL 1.8 × 10^10^/mL percutaneous injection	42 days	Reduced the size of bladder tumors and induced apoptosis of tumor cells	Chang et al., (2014) [3]
BALB/c nude mice	0.1 mL 2 × 10^10^/mL s.c.	6 weeks	It reduced the size of pancreatic tumors and induced the apoptosis of tumor cells	Cheng et al., (2016) [41]
BALB/c nude mice	0.1 mL 2.2 × 10^10^ cells/mL s.c.	N/A	Increased sensitivity of non-small cell lung tumors to gefitinib	Zhao et al., (2016) [16]
BALB/c nude mice	0.1 mL 2.2 × 10^10^ cells/mL s.c.	6 weeks (every other day)	Induced ER stress and consequently promoted autophagy in HR-breast cancer	Xu et al., (2014) [28]
BALB/c nude mice	0.1 mL 1 × 10^10^/mL peritumor injection	2 weeks (every second day)	Reduced mass and volume of cetuximab-resistant colorectal cancer and prolonged survival	Zhang et al., (2022) [58]
BALB/c mice	1 × 10^10^/d ip	40 days	Inhibited the growth and progression of hepatocellular carcinoma and induced the apoptosis of tumor cells	Li et al., (2014) [63]
BALB/c mice	10^6^/mL s.c.	38 days	Inhibited the progression of gastric tumors	Wang et al., (2015) [31]
C57bl/6J mice	2 × 10^8^ pcs/mL 4 × 10^8^ pcs/mL 6 × 10^8^ pcs/mL ip	7–10 weeks (every fifth day)	Inhibited bladder tumor growth, reduced its invasiveness, and promoted tumor cell apoptosis	Huang et al., (2022) [55]
C57BL/6 wild-type and nude mice C57BL/6 TLR42/2, and TLR22/2 mice	10^6^ CFU/mL 10^7^ CFU/mL 10^8^ CFU/mL s.c.	Until the tumor has grown to a diameter of 12 mm or until metastases have developed	Slowed down the growth of lung tumors and prolonged the survival of mice	Zhang et al., (2014) [54]
C57BL/6 female mice	1.6–2.0 × 10^9^ 3.2–4.0 × 10^8^ 6.4–8.0 × 10^7^ CFU/mL s.c.	3 weeks (twice a week)	Stimulated immune response in the murine tumor model prolonged mouse survival and reduced tumor growth.	Li et al., (2015) [57]
Orthotopic mice	1 × 10⁹ CFU/mL	3 weeks (once a week)	Induction of effector T cells and stimulation of immune-suppressing mechanisms	Wang et al., (2022) [61]
Nude mice	1.6–2 × 10^9^ cells/mL s.c.	6 weeks	Inhibited breast tumor growth and induced apoptosis of tumor cells resistant to doxorubicin	Wei et al., (2016) [44]
Wistar rats	1 × 10^6^/mL iv	10 weeks (every third day)	Inhibited the growth of bladder tumors, reduced invasiveness	Liu et al., (2017) [56]

ip—intraperitoneal injection; s.c.—subcutaneous injection.

## 4. Anti-Tumor Potential of PA-MSHA in Humans: Past and Current Clinical Studies

Similar to the results seen in in vivo studies, administration of PA-MSHA has been reported to modulate the immune response in hosts by interacting with the T-cell response, promoting the maturation of dendritic cells, and increasing the polarization of pro-inflammatory M1 macrophages. These immune-related effects were also confirmed in clinical trials, where subcutaneous injections of PA-MSHA enhanced immunity in cancer patients [64]. The first clinical data were reported in 1999 when Li and colleagues applied PA-MSHA to lung cancer patients for 10 weeks as an adjunct to chemotherapy. They found that PA-MSHA-treated patients had lower infection rates than control subjects (15.91% vs. 40%) and showed enhanced immune function. Patients receiving PA-MSHA showed a significantly higher level of cytotoxic natural killer (NK) cell activity and pro-inflammatory IL-2, as well as a higher CD4-to-CD8 effector T cell ratio than those in the control group [65]. The following year, the same research group reported the results of another clinical study in which the addition of PA-MSHA improved the efficacy of chemotherapy from 69.77% to 95.56% in patients with malignant lymphoma and from 42.22% to 59.09% in lung cancer patients [66]. Similarly, PA-MSHA injection showed positive anti-tumor effects in 30 breast cancer patients who underwent the TAC (docetaxel, doxorubicin, and cyclophosphamide) protocol. Following two rounds of chemotherapy, individuals treated with PA-MSHA exhibited a significant reduction in tumor size, rendering the tumors suitable candidates for resection in surgery [67]. Even in severe cases of metastatic HER-2 negative breast cancer, patients treated with a combination of capecitabine and PA-MSHA experienced longer progression-free survival time (8.2 months vs. 4 months) and improved overall survival (25.4 months vs. 16.4 months). Moreover, researchers noted that moderate immune-related adverse events such as fever or skin induration at the injection site were in correlation with better treatment outcomes, indicating dose-dependent effects of PA-MSHA [32]. In 2023, Gong and colleagues conducted a study where 75 patients with HER-2-negative breast cancer received paclitaxel and carboplatin with or without PA-MSHA. They found that adding PA-MSHA to neoadjuvant chemotherapy could improve the clinical response of cancer patients by modulating the hosts’ immune response. PA-MSHA-treated patients had higher serum IFN-γ levels and a higher percentage of effector T cells, such as CD8+/CD4+ T cells, CD8+CD28+ T cells, as well as NK cells, and decreased serum IL-4 levels, indicating the active state of the immune system. As expected, patients with immune-related adverse events benefited more from the PA-MSHA treatment than patients not experiencing any immune-related effects. Hence, scientists have proposed the pivotal role of the immune system stimulated by PA-MSHA in the context of cancer treatment [29].

Finally, Zhang et al., (2017) performed a clinical study that included 20 patients who suffered from lung, ovarian, colon, rectal, and esophageal cancer. Patients were divided into two groups: the first receiving chemotherapy plus CIK cells derived from patients’ own cord blood, and the second receiving chemotherapy plus PA-MSHA treated CIK cells. They observed that PA-MSHA stimulated proliferation of CIK cells and decreased the expression of inhibitory cell surface markers, Tim-3 and PD-1. Moreover, PA-MSHA-treated CIK cells secreted more proinflammatory cytokines, such as IFNγ, indicating stronger activation of the immune system [68].

Summarized results of clinical studies are shown in Table 2.

## 5. PA-MSHA-Induced Side Effects

While the in-depth toxicological profile of PA-MSHA has not been investigated yet, data from in vivo and clinical studies suggest that it has a good safety profile. It was reported that PA-MSHA can cause non-hematologic toxic side effects such as skin irritation in 44.3%, hand-foot syndrome in 32%, fever in 29.9%, or rash in 5.2% of treated patients with HER-2 negative metastatic breast cancer when applied in combination with taxol. Moreover, it induced mild and moderate hematologic toxicity, while 5% of treated patients experienced severe neutropenia, leukopenia, and thrombocytopenia [32]. Gong et al., (2023) described the same adverse effects in cancer patients and reported that the most common side effects seen in PA-MSHA-treated patients were mild to moderate skin induration at the injection site, rash, and fever. However, 64.9% of patients experienced at least one grade 3 or 4 side effect (the severity of the side effects was graded from 1 to 4). The most common severe side effects were neutropenia and leukopenia, found in 59.5% and 40.5% of patients, respectively. Apart from hematologic toxicity, peripheral neuropathy was seen as a severe side effect in 5.4% of patients. Moreover, they noted that the severity of the mentioned side effects positively correlated with the treatment outcome, confirming the role of the immune system in PA-MSHA-induced cancer regression [29]. All reported side effects were linked to the induced release of pro-inflammatory cytokines, mainly IFNγ, and immune system stimulation [33]. On the other hand, Chen et al., (2009) reported that PA-MSHA reduced the incidence and severity of TAC-mediated side effects such as accumulation of subcutaneous fluid, skin flap necrosis, and infection in patients with breast cancer [67], showing the need for further understanding of the toxic potential of PA-MSHA. More studies are required to confirm the optimal, safe dose of PA-MSHA.

## 6. Future Directions

Available literature suggests the anti-tumor potential of PA-MSHA as an adjuvant to chemotherapy in various cancer types. It was noted that PA-MSHA injections stimulate an inflammatory response in cancer patients, straightening the host’s immune system to fight cancer through various mechanisms. However, overstimulation of the immune response can cause side effects such as fever or irritation at the application site. Therefore, future research should be directed toward further understanding of PA-MSHA interactions with the immune system, optimization of the dose used in clinical settings, and highlighting the patients that should benefit the most from PA-MSHA. Next, it is yet to be addressed if PA-MSHA could be used as a monotherapy in cancer treatment. Three arguments were suggested that favor the use of monotherapy: lower cost, low risk of resistance, and unpredictable benefits of combination therapy [69]. Moreover, limited data about PA-MSHA-induced toxicity indicates the need for further exploration and in-depth understanding of its toxic potential. Current data shows that immune-related side effects of PA-MSHA can be associated with hematological toxicity and cause neutropenia or leukopenia [29,32]. Thus, the next step of PA-MSHA research should aim to decipher the toxicity of PA-MSHA.

## 7. Conclusions

The development of immune checkpoint inhibitors highlighted the importance of the immune system in fighting cancer, bringing immune-modulating therapies into the focus of cancer research. Accordingly, bacteria-mediated cancer therapy, including PA-MSHA, has emerged as a novel antineoplastic strategy. Even though the anti-cancer properties of PA-MSHA have been extensively studied in the past, only limited data about its positive and negative effects is available. For example, it was demonstrated that PA-MSHA can be a useful adjuvant to chemotherapy by promoting tumor suppression in a dose-dependent manner in various types of cancer, such as lung and breast cancer, as well as in lymphoma. Inhibition of the cell cycle, induction of apoptosis, activation of the immune system, and interference with key signaling pathways such as EGFR, Nrf2/p62, and miR-7-5p/Akt/Wnt-β-catenin were some of the proposed underlying anti-tumor mechanisms mediated by PA-MSHA. Based on the available data, it can be suggested that PA-MSHA activates signaling pathways linked to inflammation and, consequently, triggers the apoptosis of cancer cells. However, when injected into patients, the activation of the inflammatory response may intertwine with the activation of the host’s defensive anti-inflammatory mechanisms, potentially mitigating the effects of PA-MSHA. These unanswered questions require further research to explore and understand the full potential of PA-MSHA in cancer treatment, as well as its optimal doses. A thorough evaluation of the safety profile of PA-MSHA is imperative, considering its potential dual role in cancer treatment—where it may aid by stimulating the immune response but could also elicit side effects like fever; skin irritation; and pain at the application site. Consequently, further in vitro and in vivo studies are essential to elucidate the benefit-to-risk ratio associated with PA-MSHA use in cancer patients and to establish a comprehensive safety profile for its therapeutic application.

## Figures and Tables

**Figure 1 cancers-16-00524-f001:**
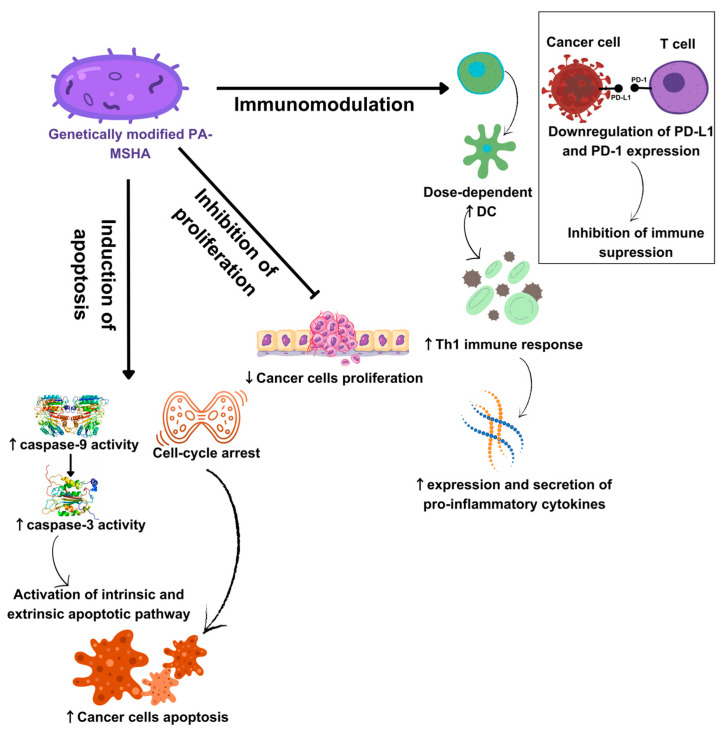
Anti-tumor mechanisms of PA-MSHA: PA-MSHA mediates tumor apoptosis by inducing intrinsic and extrinsic apoptotic pathways and via dysregulation of cancer cells’ cycle. PA-MSHA inhibits cancer cell proliferation. PA-MSHA modulates immune response by interacting with DC and activating Th1 immune response. PA-MSHA downregulates the expression of PD-1 on immune cells and PD-L1 on cancer cells, thus preventing immune suppression. DC-dendritic cells; PD-1—programmed death-1 receptor; PD-L1—programmed death-ligand 1; ↓—decreased activity; ↑—increased activity.

**Table 2 cancers-16-00524-t002:** Anti-tumor effects of PA-MSHA: evidence from clinical studies.

Treatment Doses	Duration	Effects	References
PA-MSHA 1 mL s.c. plus chemotherapy	Total: 10 weeks; 1st week: 0.5 mL of PA-MSHA was injected on the day 1, 3, 5. From 2nd to 10th week: 0.8 mL was injected at day 5	PA-MSHA increases T cell proliferation and the production of pro-inflammatory cytokines in lung cancer patients to prevent super-infection.	Li et al., (1999) [65]
PA-MSHA 1 mL s.c. plus chemotherapy		PA-MSHA improves the effectiveness of treatment of lymphoma and lung cancer by modulating patients’ immune response.	Li et al., (2000) [66]
PA-MSHA 1 mL s.c. in addition to the TAC * scheme	Cycle: 3 weeks Total: 2–4 cycles	PA-MSHA enhanced therapeutic response of breast carcinoma patients treated with TAC scheme.	Chen et al., (2009) [67]
PA-MSHA 1 mL s.c. in addition to 1000 mg/m^2^ of capecitabine twice a day	Cycle: every other day for 2 weeks, 1 week off. Total: 2–4 cycles	PA-MSHA and capecitabine possess superior clinical efficacy in patients with metastatic breast cancer compared to either treatment alone.	Lv et al., (2015) [32]
CIK cells treated with 4.5 × 10^6^, 9 × 10^6^, 13.5 × 10^6^, 18 × 10^6^, and 22.5 × 10^6^ CFU/mL of PA-MSHA and combined with chemotherapy	Cycle: different chemotherapy protocols. CIK cells treated with PA-MSHA were administered 1 day after the chemotherapy. Total: 2–6 cycles	PA-MSHA stimulated the proliferation of CIK cells in a dose-dependent manner. CIK cells acquired a more cytotoxic phenotype: increased production of IFN-γ, IL-2, and CD107a.	Zhang et al., (2017) [68]
1 mL every other day (0.5 mL on the first day) from the first day of neoadjuvant chemotherapy (paclitaxel and carboplatin) until 3 days before surgery	Cycle: 4 weeks Total: 4 cycles of neoadjuvant chemotherapy	The addition of PA-MSHA to neoadjuvant chemotherapy in HER2-negative breast cancer improves the tumor’s clinical response. Patients with immune-related adverse events could benefit more from the PA-MSHA treatment.	Gong et al., (2023) [29]

* TAC scheme: docetaxel, doxorubicin, and cyclophosphamide.

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
