# Peer review of "Trends in Anti-Tumor Effects of Pseudomonas aeruginosa Mannose-Sensitive-Hemagglutinin (PA-MSHA): An Overview of Positive and Negative Effects"

_cancers, 2024, doi:10.3390/cancers16030524_

Round 1

Reviewer 1 Report

Comments and Suggestions for Authors

The authors have comprehensively explored the literature and background surrounding PA-MSHA in cancer treatment. Further research is warranted to investigate other bacterial products for their potential as anti-cancer therapeutic agents. There are a few aspects that need to be revised before the publication.

1.    Consider adjusting the title to reflect the limited data available on the toxicological pattern of Pseudomonas aeruginosa-mannose sensitive hemagglutinin (PA-MSHA).

2.    Enhance the conclusion to further emphasize PA-MSHA's applications and future prospects in cancer therapy.

Comments on the Quality of English Language

The authors have extensively reviewed the literature and background concerning PA-MSHA in cancer treatment. Some grammatical and typographical errors were identified, and I kindly ask the authors to rectify them accordingly.

(i)            Line 102: heath-killed?

(ii)          Line 147: reviled ?

(iii)         Line 153: respectfully?

(iv)         Line 157: underlaying?

(v)          Line 160: the sentence seems unclear “The cells growth inhibition was the most prominent after 72h”

(vi)         Line 255 “snown”

(vii)        Line 298 the sentence seems unclear

(viii)       Line 394 “Positevely”

Author Response

1. Thank you very much for your comments. We have slightly changed the title as there is limited data about its toxicity.  The new title is: Current trends of in anti-tumor effects of Pseudomonas aeruginosa mannose-sensitive-hemagglutinin (PA-MSHA): an overview of positive and negative effects. The new title is more suitable as the review does cover both the positive and negative effects of PA-MSHA but also notes that the toxicity data is limited and should be further investigated.

2. The conclusion was rewritten and updated. We also added a paragraph about future research related to the use of PA-MSHA in cancer therapy.

3. All grammatical and typographical errors were corrected according to the instructions. Grammar was checked also.

Reviewer 2 Report

Comments and Suggestions for Authors

This review summarizes the current knowledge obtained from in vitro, in vivo, and clinical studies, regarding the use of PA-MSHA in cancer treatment to further elucidate its pharmacological and toxicological properties. This review article is well-written and acceptable.

Author Response

Thank you very much for your comments, we appreciate it.

Reviewer 3 Report

Comments and Suggestions for Authors

The review titled "Current trends in anti-tumor effects of Pseudomonas aeruginosa mannose-sensitive-hemagglutinin (PA-MSHA): pharmacological and toxicological point of view" addresses the limitations of traditional cancer treatments and introduces the promising alternative of bacteria-mediated cancer therapy using PA-MSHA.

The purpose is clear, and the inclusion of evidence from in vitro, in vivo, and clinical studies enhances the credibility of the presented information. However, to further strengthen the review, a more detailed exploration of the proposed mechanisms of action and the incorporation of quantitative data on the efficacy of PA-MSHA would be beneficial.

 Additionally, a more comprehensive discussion of the toxicological properties, including the frequency and severity of adverse reactions, could provide a more nuanced understanding. Finally, considering the inclusion of a section on future directions or areas requiring further research would contribute to the completeness of the review.

The Authors should increase the number of references for this work as possible

You have to add to the abstract the database sources

The references should be written in number style not authors and date

To improve the introduction, it is worth to add the main cancer therapy targets and you can use the following references “Highlights on Specific Biological Targets; Cyclin-Dependent Kinases, Epidermal Growth Factor Receptors, Ras Protein, and Cancer Stem Cells in Anticancer Drug Development” and “Biomolecules 2022, 12, 1843.” That will make the frequency of the story more attractive.

All bacterial strains should be written in italic style through the manuscript

One important note that you could not use a unit like cell/ml for the IC50, because the IC50 means inhibitory concentrations, and I think the concentrations should be in related units (like µg/mL)

In figure 1 you have to improve the font style with better resolution

The paragraphs from lines 131-138 and 222-250 were written by AI programe it should be re-phrased accordingly

The Conclusion should be improved

Best wishes

Author Response

1. Thank you for your comments. We have explained the anti-cancer mechanisms of PA-MSHA in more detail and added quantitative data about its efficacy where appropriate (lines 418, 419 in 'manuscript marked with changes').
2. We have also included more details about toxicological data from clinical studies which describe the frequency and severity of adverse reactions.
3. The section 'future directions' was incorporated into the manuscript.
4. We have added 12 new references to the review.
5. We added database sources in the abstract. We used Pubmed, Google Scholar, and ClinicalTrials.gov for literature research.
6. References were changed to numerical style.
7. We added the main cancer therapy targets into the introduction and cited both references (Highlights on Specific Biological Targets; Cyclin-Dependent Kinases, Epidermal Growth Factor Receptors, Ras Protein, and Cancer Stem Cells in Anticancer Drug Development” and “Biomolecules 2022, 12, 1843).
8. We have checked and made all bacterial strains in italics.
9. Thank you for noticing. However, as this is the review paper, we used concentrations and units in a way they were presented in the published papers we described and cited.
10. The font and resolution of Figure 1 were improved.
11. The paragraphs from lines 131-138 and 222-250 were rewritten as suggested.
12. Conclusion was improved. 

Round 2

Reviewer 3 Report

Comments and Suggestions for Authors

all asked comments and recommendation were solved and improved, the manuscript was well improved